# Magnetic Properties of a Solid Solution $Fe_{1-x}Ag_xCr_2S_4$ (0 < x < 0.2)

**Galina G. Shabunina, Elena V. Busheva, Pavel N. Vasilev *** and **Nikolay N. Efimov ***

N. S. Kurnakov Institute of General and Inorganic Chemistry of Russian Academy of Sciences, 31 Leninsky Prosp., 119991 Moscow, Russia
* Correspondence: anubisvas@gmail.com (P.N.V.); nnefimov@narod.ru (N.N.E.)

**Abstract:** The magnetic properties of the $Fe_{1-x}Ag_xCr_2S_4$ (0 < x < 0.2) solid solution were investigated in the temperature range 4–300 K in a DC field of 0.1 and 45 kOe. $Fe_{1-x}Ag_xCr_2S_4$ is characterized by the transition temperature from the paramagnetic to the ferromagnetic state ($T_c$) and the irreversibility temperature ($T_{irr}$). The replacement of iron with silver ions in $Fe_{1-x}Ag_xCr_2S_4$ leads to an increase in the Curie temperatures from 185 K (x = 0) to 216 K (x = 0.2) and $T_{cusp}$ from 45 K (x = 0) to 125 K (x = 0.2). A section of the $Fe_{1-x}Ag_xCr_2S_4$ magnetic phase diagram in the region under study has been constructed. The diagram reveals the following regions: paramagnetic, ferromagnetic, region of conditionally recurrent spin-glass (SG), and below 10 K a region associated with orbital ordering.

**Keywords:** magnetic semiconductor; paramagnetism; chalcogenide spinel

## 1. Introduction

The work is dedicated to the study of the magnetic properties of spinel solid solutions based on a silver-doped iron dichromium tetrasulfide $FeCr_2S_4$. Interest in the effect of the replacement of iron ions by silver ions is due to the possibility of increasing the magnetic ordering temperature. At the same time, the interest of researchers in the Cr-based spinels, which have found to exhibit a unique combination of magnetic, semiconductor, and striction properties [1,2], and also have an unusual magnetic behavior at low temperatures, such as colossal magnetoresistance (CMR), has recently increased [3–5].

According to its properties, iron dichromium tetrasulfide ($FeCr_2S_4$) is a ferrimagnet ($T_C$ = 177–185 K) with a normal spinel structure, space group Fd3m. The magnetic moments of the $Fe^{2+}$ and $Cr^{3+}$ ions in $FeCr_2S_4$ at a temperature T = 4.2 K, according to the data of [6,7], are 4.2 $\mu B$ and 2.9 $\mu B$, respectively. The value of the magnetic moment per $FeCr_2S_4$ molecule, equal to 1.8 $\mu B$, is in good agreement with the calculated moment for a simple Neel configuration [8–11]. The temperature dependence of the reciprocal molar susceptibility of the compound confirms the ferrimagnetic nature of the ordering. In this case, the Curie–Weiss law is fulfilled with the asymptotic Curie temperature θ = −260 ± 40 K [10].

At temperatures below 100 K, $FeCr_2S_4$ exhibits behavior similar to that of a spin glass: in a weak magnetic field. The compound exhibits the irreversibility of the zero field cooling (ZFC) and field cooling (FC) magnetization curves and parabolic anomalies in the region of T = 70 K due to a sharp increase in magnetic anisotropy. With a further decrease in temperature, a stepwise decrease and a subsequent rise in the ZFC and FC magnetization curves are revealed, which is associated with the presence of a structural transition at T = 9 K due to orbital ordering [12–18]. Taking into account the listed features in the studied system of solid solutions, one should apparently expect manifestations of unusual magnetic properties. The $FeCr_2S_4$ chalcospinel is a classical magnetic semiconductor, but as it has now been established, being a normal Heisenberg magnet, this compound, as it approaches absolute saturation in the region of helium temperatures, begins to show signs of instability, similar to a large extent to spin-glass or superparamagnetic behavior [19–21]. The nature of such magnetic behavior has not yet been fully elucidated.

The existence of a phase transition in $FeCr_2S_4$ at a temperature close to 60 K has been confirmed by transmission electron microscopy and measurements of *dc*- and *ac*-magnetic susceptibility [3,17,22].

Muon spin relaxation spectroscopy has shown that the ferrimagnetic structure of $FeCr_2S_4$ transforms at T~60 K into an incommensurate, nonsinusoidal spin structure, remaining stable up to temperatures of the orbital ordering state [23]. To interpret the properties of $FeCr_2S_4$, different models have been developed, related both to the transition in the symmetry of the structure and to the reentrant behavior of spin-glass or magnetic domains reorganisation.

Previously, our group synthesized samples of $Fe_{1-x}Ag_xCr_2S_4$ solid solutions from elements by the method of solid-phase reactions, which have a limited homogeneity region equal to x = 0.22 [24]. According to XRD data, samples with $0 < x \leq 0.2$ were single-phase. With an increase in the silver content in the sample, the unit cell parameter increased in accordance with Vegard's law (Figures S1 and S2).

It was found that the iron substitution by silver resulted to an increase in the Curie temperature by about 30 K (from 185 K (x = 0) to 214 K (x = 0.20) [24]. In this case, the solid solution based on iron dichromium tetrasulfide remained a ferrimagnet with a state characteristic of it at low temperatures, close to the state of a spin glass.

Also, in [25] for $Fe_{1-x}Ag_xCr_2S_4$ (x = 0.05–0.15), we measured the temperature dependences of the real ($\chi'$) and imaginary ($\chi''$) parts of the *ac*-magnetic susceptibility at frequencies of 10–10,000 Hz with amplitudes $H_{AC}$ = 1 Oe and 15 Oe, which confirmed the nature and temperatures of magnetic transitions established in [24]. Namely, polycrystalline $Fe_{1-x}Ag_xCr_2S_4$ was characterized not only by an increase in the Curie temperature ($T_C$), but also by an increase in the temperature of the spin-glass state ($T_f$), depending on the silver concentration. It is assumed that an increase in the transition temperatures with an increase in the silver concentration may be due to the effect of diamagnetic dilution.

The presence of a cusp near $\approx$ 50 K, associated with a low-temperature structural anomaly for the $Fe_{1-x}Ag_xCr_2S_4$ solid solution, was confirmed by measuring the temperature dependence of the imaginary part of the *ac*-magnetic susceptibility $\chi''$ (T), with an increase in amplitude up to 15 Oe. The discrepancy in the behavior of the *ac*-susceptibility can be associated with the manifestation of domain wall pinning. This assumption has been justified by the application of an alternating field with a higher amplitude $H_{AC}$ = 15 Oe, which overcomes the pinning effect. Also, for $Fe_{1-x}Ag_xCr_2S_4$, a transition was found at a temperature of $T_{OO}$ = 9–13 K, which is considered the temperature of long-range orbital ordering caused by the Jahn–Teller effect.

Taking into account the above, the $Fe_{1-x}Ag_xCr_2S_4$ solid solution is a non-trivial object for magnetic property studying. This primarily applies to the maximum temperature used in magnetic measurements, as in our case, it is limited by the technical capabilities of the measuring equipment, as well as by the thermal stability of the solid solution under study.

The accuracy of determining the paramagnetic Curie temperature $\theta_P$ depends on the length of the high-temperature (above $T_c$) segment $\Delta T$ used to cut off the constant $\theta_P$ by extrapolating the $\chi^{-1}$ versus T dependence. The possibility to somewhat refine the characteristics in paramagnetic range was additionally provided by the data on the measurement in [10] of the magnetic susceptibility of $FeCr_2S_4$ at high temperatures up to 1200 K ($\theta_P$ = −260 K), which made it possible to use them in this work as reference [3,22].

In present study our first priority was to measure and interpret the magnetic properties of the $Fe_{1-x}Ag_xCr_2S_4$ solid samples in order to better understand the nature and mechanisms of the magnetic interactions in this solid solution.

## 2. Results and Discussion

The Figures 1a, 2a, 3a and 4a show the temperature dependences of the magnetization $M(T)_{ZFC}$ and $M(T)_{FC}$ of a $Fe_{1-x}Ag_xCr_2S_4$ solid solution with x = 0.05, 0.10, 0.15, and 0.20 in the range from room to helium temperature in a weak magnetic field H = 100 Oe. As can be seen from these figures, the dependences $M(T)_{ZFC}$ and $M(T)_{FC}$ of the samples

are characterized by two maxima of the temperature derivative of magnetization, the first of which corresponds to the transition from a paramagnetic state to a ferrimagnetic long-range ordered phase with Curie temperatures $T_C$ = 185–216 K (see Table 1). The Curie temperature for samples of same composition was in a good agreement with each other, regardless of whether it was determined from $M(T)_{ZFC}$ or $M(T)_{FC}$ and rising almost linearly with composition of Ag and reaches the value of 216 K (x = 0.2) (Figure 5). Effects corresponding to the probable ferrimagnet–spin glass transition ($T_{cusp}$) were also present in both $M(T)_{ZFC}$ and $M(T)_{FC}$ dependences. The temperature of the cusp increased with increasing silver concentration.

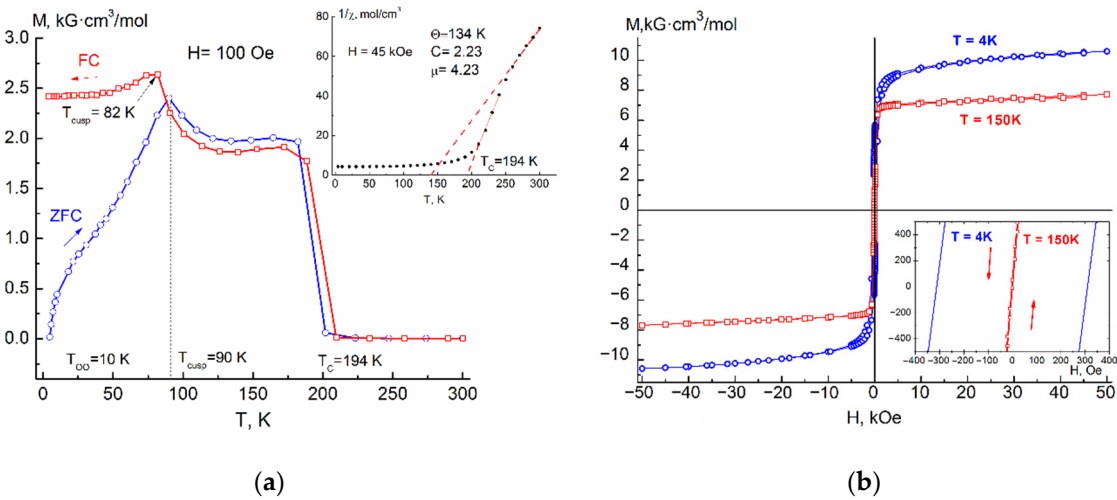

(a)                                                    (b)

**Figure 1.** (**a**) Dependences of the M vs. T of FC and ZFC ($M(T)_{ZFC}$ or $M(T)_{FC}$) magnetization for $Fe_{0.95}Ag_{0.05}Cr_2S_4$ in a 100 Oe field. Inset: dependence of the inverse magnetic susceptibility versus temperature for this composition. (**b**) Dependences of the magnetization versus magnetic field for $Fe_{0.95}Ag_{0.05}Cr_2S_4$ at 4 and 150 K.

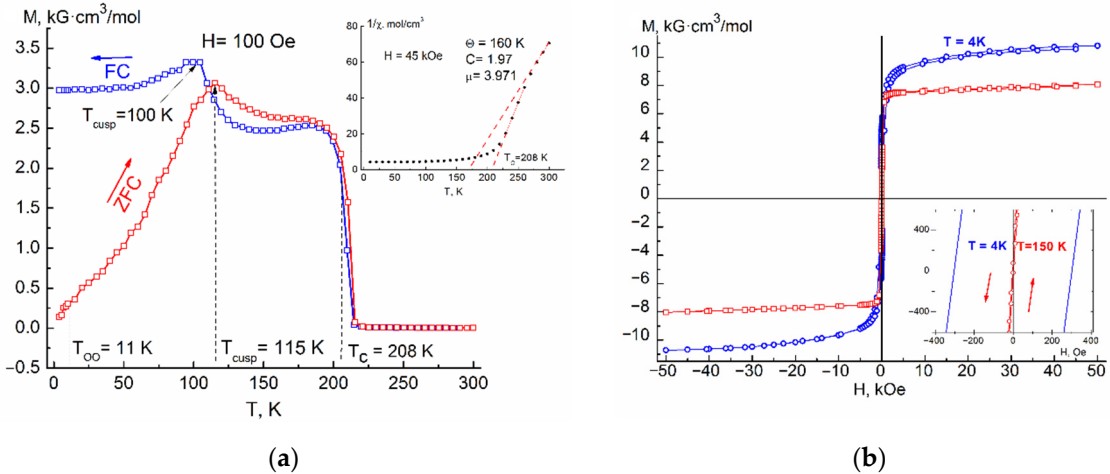

(a)                                                    (b)

**Figure 2.** (**a**) Dependences of the M vs. T of FC and ZFC magnetization for $Fe_{0.90}Ag_{0.10}Cr_2S_4$ in a 100 Oe field. Inset: dependence of the inverse magnetic susceptibility versus temperature for this composition. (**b**) Dependences of the magnetization versus magnetic field for $Fe_{0.90}Ag_{0.10}Cr_2S_4$ at 4 and 150 K.

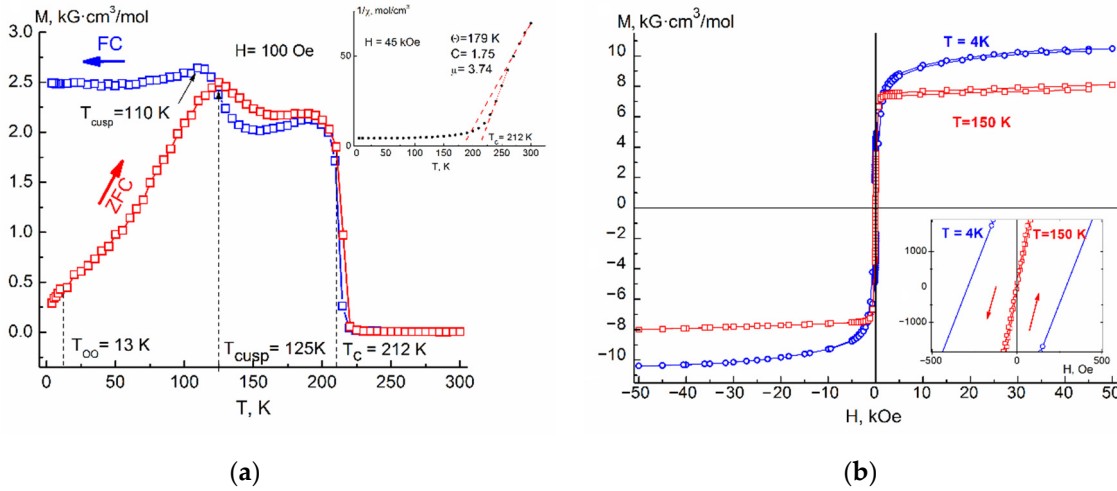

**Figure 3.** (**a**) Dependences of the M vs. T of FC and ZFC magnetization for $Fe_{0.85}Ag_{0.15}Cr_2S_4$ in a 100 Oe field. Inset: dependence of the inverse magnetic susceptibility versus temperature for this composition. (**b**) Dependences of the magnetization versus magnetic field for $Fe_{0.85}Ag_{0.15}Cr_2S_4$ at 4 and 150 K.

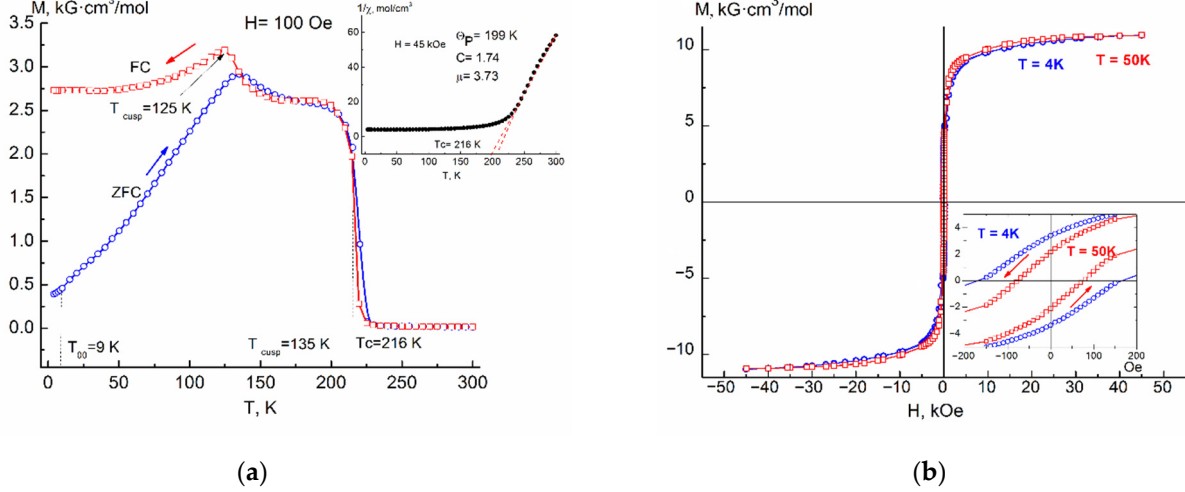

**Figure 4.** (**a**) Dependences of the M vs. T of FC and ZFC magnetization for $Fe_{0.80}Ag_{0.20}Cr_2S_4$ in a 100 Oe field. Inset: dependence of the inverse magnetic susceptibility versus temperature for this composition. (**b**) Field dependences of the magnetization for $Fe_{0.80}Ag_{0.20}Cr_2S_4$ at 4 and 150 K.

**Table 1.** Magnetic parameters of $Fe_{1-x}Ag_xCr_2S_4$ solid solutions.

| Composition $x_{Ag}$ | $a,$ Å | $T_C$, K | $T_{cusp\ FC}$, K | $T_{cusp\ ZFC}$, K | $T_{OO}$, K | $\Theta_P,$ K | C | $\mu_{eff}$, $\mu_B$ |
|---|---|---|---|---|---|---|---|---|
| 1 | 2 | 3 | 4 | 5 | 6 | 7 | 8 | 9 |
| 0 | 9.990 | 185 | 45 | 50 | 10 | −260 | 3.96 | 5.63 |
| 0.05 | 9.993 | 194 | 82 | 90 | 10 | 134 | 2.23 | 4.23 |
| 0.1 | 9.996 | 208 | 100 | 115 | 11 | 160 | 1.97 | 3.97 |
| 0.15 | 10.002 | 212 | 110 | 125 | 13 | 179 | 1.75 | 3.74 |
| 0.2 | 10.004 | 216 | 125 | 135 | 9 | 199 | 1.74 | 3.73 |

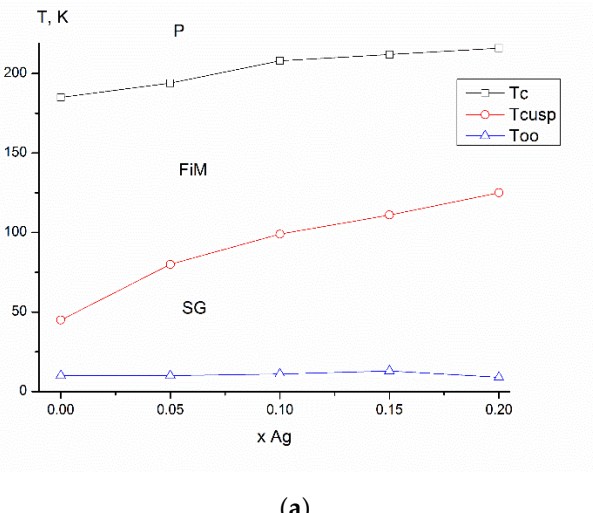
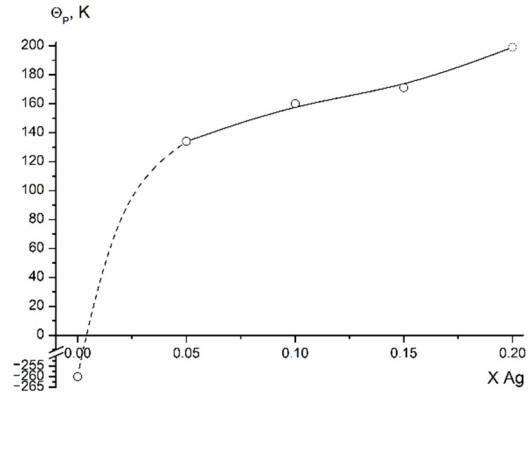

(**a**)                                              (**b**)

**Figure 5.** (**a**) Section of the magnetic phase diagram for $Fe_{1-x}Ag_xCr_2S_4$. (**b**) Concentration dependence of the paramagnetic Curie temperature for $Fe_{1-x}Ag_xCr_2S_4$.

As can be seen from Figures 1a, 2a, 3a and 4a, the $Fe_{1-x}Ag_xCr_2S_4$ solid solution is characterized by the transition temperature from the paramagnetic to the ferrimagnetic state ($T_C$), and the irreversibility temperature ($T_{irr}$) (the temperatures at which the curves $M(T)_{ZFC}$ and $M(T)_{FC}$ begin to diverge), which practically similar with the $T_c$ of the sample. The divergence of the $M(T)_{ZFC}$ and $M(T)_{FC}$ curves is observed for temperatures well below the Curie point. In the $M(T)_{FC}$ case, the spins are blocked in the direction depending on the orienting external magnetic field which results in higher magnetization values at low temperatures.

Unlike $T_C$, the temperature of irreversibility ($T_{irr}$) is not a characteristic parameter of the material. Its value may depend on the deviation from the stoichiometry of the sample, its crystalline state (single- or polycrystalline), as well as the presence of uncontrollable impurities [6]. Due to the fact that in all the $Fe_{1-x}Ag_xCr_2S_4$ samples studied in this work, the values of the Curie temperatures $T_C$ and irreversibility $T_{irr}$ practically coincide with each other, it can be assumed that this phenomenon has a common nature. The most probable reasons of such irreversibility are the large value of the magnetocrystalline anisotropy constant $K_1$ and the pinning of magnetic domain walls. Thus, for $FeCr_2S_4$, the $K_1$ value is one of the largest among spinels. In the magnetically ordered compound $FeCr_2S_4$ [17] it has a positive value and at $T = 4.2$ K it is equal to $K_1 = 3–6 \times 10^6$ erg/cm$^3$ [3].

Note that the $FeCr_2S_4$ magnetization curves saturate only in high fields ($H \approx 5$ T) and low temperatures ($T \approx 4$ K) [9,19]. The need for strong fields to saturate the magnetization at low temperatures is explained by a strong increase in the magnetocrystalline anisotropy of the compound [5].

The introduction of nonmagnetic silver ions into $FeCr_2S_4$ creates additional pinning centers, which broaden the magnetic transition caused by a dramatic increase in the magnetocrystalline anisotropy at low-temperature.

In Figures 1a, 2a, 3a and 4a, attention is drawn to the gentle maxima (cusps) on the curves $M(T)_{ZFC}$ and $M(T)_{FC}$, which lie (depending on the concentration of introduced silver) at $T = 45–125$ K (see Table 1, column 4). The origin of these maxima, as shown in [8], is associated with the original iron thiochromite ($FeCr_2S_4$), the extreme composition of the $Fe_{1-x}Ag_xCr_2S_4$ system. According to the data of [3], the temperature dependence of the magnetization in undoped $FeCr_2S_4$ has a number of traits that are more characteristic of cluster spin glasses or superparamagnets. However, in the case of $Fe_{1-x}Ag_xCr_2S_4$, we are dealing with a more complex phenomenon that has features of a cluster spin glass. For the purposes of this study, all compositions can be considered as conditionally spin-glass

ones, since elucidation of low-temperature anomalies origins in $FeCr_2S_4$ and its diluted compounds is outside the scope of this study.

Confirmation that the presence of effects such as cusp or spin glass at T = 45–125 K in the samples is associated with $FeCr_2S_4$ is an increasing temperature of these effects with an increase in the silver concentration, as well as their gradual smearing, indicating that with increasing x there is a weakening antiferromagnetic A-S-B exchange interaction.

As can be seen from Figures 1a, 2a, 3a and 4a, as the temperature decreases below the Curie point in the $Fe_{1-x}Ag_xCr_2S_4$ sample, ZFC and FC magnetization curves immediately shows an irreversible behavior, then an anomalous cusp appears at $T_{cusp}$, caused by a significant increase in magnetic anisotropy and subsequent spin reorientation of the domains. With a further decrease in the temperature of the sample, one more anomalous transition is revealed in the region of T ≈ 10 K, which has a stepwise motif, a decrease (ZFC) or an increase (FC) of the magnetizations associated with the onset of orbital ordering at a $T_{OO}$ temperature.

Figures 1b, 2b, 3b and 4b show isothermal field dependences of the magnetization of the solid solution $Fe_{1-x}Ag_xCr_2S_4$ (x = 0.05–0.20), measured up to H = 5 T, for various temperatures below $T_C$. The shape of these dependences is characteristic of a ferrimagnet with parameters presented in Table 2.

**Table 2.** Characteristics of hysteresis loops of $Fe_{1-x}Ag_xCr_2S_4$ solid solutions at different temperatures.

| x Ag | $H_C$, Oe $_{T = 4 K}$ | $H_C$, Oe $_{T = 100 K}$ | $H_C$, Oe $_{T = 150 K}$ | $M_R$, G·cm³/mol | $M_{50kOe}$, G·cm³/mol | $\mu_S$, µB |
|------|------|------|------|------|------|------|
| 1 | 2 | 3 | 4 | 5 | 6 | 7 |
| 0 | 432 | 45 | 1 | 5235 | 9886 | 1.77 |
| 0.05 | 311 | 32 | 2 | 4491 | 10,592 | 1.89 |
| 0.10 | 302 | 33 | 4 | 4598 | 10,808 | 1.93 |
| 0.15 | 289 | 34 | 4 | 3830 | 10,487 | 1.88 |
| 0.20 | 166 | | | 3511 | 10,229 | 1.83 |

Figure 5a shows a section of the magnetic phase diagram for $Fe_{1-x}Ag_xCr_2S_4$ in the region 0 < x < 0.20, built on the basis of the measured magnetic properties summarized in the Table 1. The diagram shows the following regions: paramagnetic, ferrimagnetic, recurrent spin glass region, and region associated with orbital ordering.

The insets to Figures 1a, 2a, 3a and 4a show the temperature dependences of the inverse molar magnetic susceptibility $\chi^{-1}(T)$ of the $Fe_{1-x}Ag_xCr_2S_4$ (x = 0.05–0.20) in a magnetic field of H = 4.5 T. The shape of the curves is characteristic for the ferrimagnets.

Investigation of the area in the high temperature region available to us for measurements, made it possible to determine the paramagnetic Curie temperature $\Theta_P$, the Curie-Weiss constant C, and the effective magnetic moment $\mu_{eff}$ (Table 1). By extrapolation of the straight line constructed from the points after the break of the curve $\chi^{-1}(T)$, the values of the Curie temperature (Tc) were obtained, which coincide with Tc obtained from measurements in a weak magnetic field (Table 1, column 3).

With an increase in the silver content in the tetrahedral sublattice and a corresponding decrease in the iron content, the Curie temperatures increase as it can be seen from Figure 5. In the case of $FeCr_2S_4$ doped by diamagnetic Ag ions, the paramagnetic Curie temperature (Figure 5b) change the sign from negative to positive which corresponds to a change in the prevailing magnetic interactions in the samples.

The increase in $\Theta_P$ with an increase in the silver concentration is due to the competition between positive and negative exchange interactions, accompanied by a decrease in the iron content which leads to the iron antiferromagnetic clusters downsizing.

The obtained experimental data are summarized in Table 1, where, depending on the amount of silver introduced (column 1), the unit cell parameter (2), the Curie temperature $T_C$ (3), the temperature of the cusp on the FC curves $T_{cusp}$ (4), and the orbital ordering

temperature $T_{oo}$ (5), paramagnetic Curie temperature $\Theta_P$ (6), Curie constant C (7), effective magnetic moment $\mu_{eff}$ (8).

Table 2 shows the coercive force $H_c$ (2–4) for various temperatures below Tc, the residual magnetization $M_R$ (5), the maximum measured saturation magnetization $M_{50kOe}$ in a field H = 5 T at T = 4 K (6) and the magnetic moment calculated from its saturation $\mu_S$ (7). Due to the diamagnetic substitution of $Ag^+$ ions for $Fe^{2+}$ ions, the resulting magnetic saturation moment slightly increases due to a decrease in the competition between the magnetic moments in the sublattices of the $Fe^{2+}$ and $Cr^{3+}$ ions.

## 3. Materials and Methods

The following elements were used to obtain $Fe_{1-x}Ag_xCr_2S_4$ solid solutions: Ag (99.999%), S (99.999%), Fe (99.98%) and Cr (99.8%) (Merck, Darmstadt, Germany). Synthesis was carried out by the method of solid-phase reactions in evacuated quartz ampoules at temperatures of 850–900 °C according to the scheme given in [24].

Powder X-ray diffraction study (powder XRD) was carried out on a Bruker D8 Advance diffractometer (CuK$\alpha$, $\lambda$ = 1.54 Å, Ni-filter, LYNXEYE detector, reflection geometry, Bruker, Billerica, MA, USA). All the samples studied in the present work were single-phase. X-ray diffraction patterns of the samples are shown on Figure S1.

The magnetic properties were measured using a PPMS-9 setup (Quantum Design, San Diego, CA, USA) in constant weak H = 100 Oe and strong H = 4.5 T magnetic fields. In the first case, ZFC, the sample was cooled down to the boiling point of liquid helium in the absence of a magnetic field, and then a weak measuring field was turned on and measurements were taken during heating. In the second case, FC, the measurements were carried out when the sample was cooled from room to helium temperature in a constant weak field. Field dependences of magnetization measurements were carried out up to magnetic field strengths H = 5 T. The magnetic transition temperature was found from the maximum of the derivative of the magnetization or the maximum of the cusp on the temperature dependence of the initial magnetization of the samples.

The paramagnetic Curie temperature ($\theta_P$) was determined by extrapolation of the linear parts of the temperature dependence of inverse susceptibility $\chi^{-1}$ (T). The Curie-Weiss constant C was found from the slope of the rectilinear part of the dependence $\chi^{-1}$ (T), and the effective magnetic moment per molecule was calculated by the formula $\mu_{eff} = \sqrt{8C}$ $\mu$B.

## 4. Conclusions

According to the experimental data obtained, $Fe_{1-x}Ag_xCr_2S_4$ solid solutions, which have a limited homogeneity region (0 < x < 0.2), are ferrimagnets. The $Fe_{1-x}Ag_xCr_2S_4$ solid solution is characterized by the transition temperature from the paramagnetic to the ferrimagnetic state Tc and the irreversibility temperature ($T_{irr}$) (the temperature at which the $M(T)_{ZFC}$ and $M(T)_{FC}$ curves begin to diverge), which is in a good agreement with the Curie temperature of the sample. The replacement of iron with silver ions in $Fe_{1-x}Ag_xCr_2S_4$ leads not only to an increase in the Curie temperatures (Tc) from 185 K (x = 0) to 216 K (x = 0.2), but also to an increase in $T_{cusp}$ from 45 K (x = 0) to 125 K (x = 0.2). A segment of the $Fe_{1-x}Ag_xCr_2S_4$ magnetic phase diagram in the region 0 < x < 0.20 has been constructed. The diagram reveals the following regions: paramagnetic, ferrimagnetic, recurrent spin glass region, and the region associated with orbital ordering. The temperature dependences of the reciprocal magnetic susceptibility $\chi^{-1}$ (T) of the solid solution $Fe_{1-x}Ag_xCr_2S_4$ (x = 0.05–0.20) in a strong magnetic field H = 4.5 T are given. The Curie–Weiss constant C and the effective magnetic moment per molecule are calculated.

The concentration dependence of the paramagnetic Curie temperature is constructed. In this case, the paramagnetic Curie temperature changes sign from negative to positive at small additions of silver.

**Supplementary Materials:** The following supporting information can be downloaded at: https://www.mdpi.com/article/10.3390/magnetochemistry8100112/s1, Figure S1: X-ray diffraction patterns for $Fe_{1-x}Ag_xCr_2S_4$ samples, Figure S2: Dependence of the parameter *a* of the crystal lattice on the content of Ag.

**Author Contributions:** Conceptualization, G.G.S. and N.N.E.; methodology, E.V.B.; validation, N.N.E.; formal analysis, P.N.V. and E.V.B.; investigation, N.N.E. and P.N.V.; data curation, N.N.E. and P.N.V.; writing—original draft preparation, G.G.S. and E.V.B.; writing—review and editing, N.N.E. and P.N.V.; visualization, P.N.V. and E.V.B.; supervision, N.N.E.; project administration, N.N.E.; funding acquisition, N.N.E. All authors have read and agreed to the published version of the manuscript.

**Funding:** The work was carried out within the framework of the state assignment of the IGIC RAS in the field of fundamental scientific research.

**Institutional Review Board Statement:** Not applicable.

**Informed Consent Statement:** Not applicable.

**Data Availability Statement:** Not applicable.

**Acknowledgments:** This research was performed using the equipment of the JRC PMR IGIC RAS.

**Conflicts of Interest:** The authors declare no conflict of interest.

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
