# Peer review of "Magnetic Properties of a Solid Solution Fe1−xAgxCr2S4 (0 < x < 0.2)"

_magnetochemistry, doi:10.3390/magnetochemistry8100112_

Round 1

Reviewer 1 Report

The authors describe their experimental work concerning magnetic properties of four Fe1-xAgxCr2S4
(0 < x < 0.2) solid solutions over a wide temperature range and for two values of DC bias field. The subject of the paper is within the scope of Magnetochemistry, in particular it fits the Special Issue "Ferrimagnetic Materials: State of the Art and Future Perspectives".

The paper is neatly written and it seems that it is free from errors that might hamper its publication. The authors have demonstrated a good knowledge of the subject and their efforts deserve a reward – the paper is publishable in the present form.

Please explain the choice of the DC values at which your experiments were carried out (0.1 and 45 kOe).

If possible, try to avoid “group” citations like in line 23 (the reviewer has also a tendency to write this way); try to describe in a sentence or two what are the main implications from the cited paper, for example: Ref. [1] examined the collosal magnetoresistance effect in spinels. The authors have found out that the effect is particularly pronounced for … etc. etc. Their line of reasoning was followed by Tokura et al.  [2], who examined ….

Minor remarks:

1.      Abstract line 6, use dot as decimal point separator

2.      Abstract line 13 “a” missing before “region”

3.      Reference 11, missing details on the journal and the title

Reviewer 2 Report

This paper by Efimov and co-workers describes the magnetic studies of a series of solid solution Fe1-xAgxCr2S4 (0 < x < 0.2). Though much magnetic susceptibility and magnetization of these materials were measured, several scientific and technical problems exist there. In my opinion, this manuscript is far from publication.

1.    The introduction is the shortest I have ever seen. I cannot think this short paragraph can let readers know the importance of this work.

2.    For a material, the basic characterization is its structure. I cannot see any structural information of the materials. The authors prepared the compounds, however, they claimed that the PXRD patterns can be seen in other references. How do you confirm the purity of your synthesized compounds?

3.    The authors didn’t characterize the solid solution. There is no proof to support the series compounds are solid solution, except the magnetic studies.

4.    Without the characterization of the materials, the magnetic studies presented here are questionable.

Reviewer 3 Report

Manuscript by P. N. Vasilev et al. entitled “Magnetic properties of a solid solution Fe1-xAgxCr2S4 (0 < x < 0.2)deals with a very interesting topic of magnetic properties of the Fe1-xAgxCr2S4 (0 < x < 0,2) solid solution in a chosen temperature range (4-300 K). In my opinion the topic is interesting, and I believe it will be of interest to wide audience of readers of Magnetochemistry.

However, there are some issues that should be resolved prior to article publishing:

  1. On page 1, after very brief section Introduction, authors start a new section, 2. Theoretical analysis. I do not see any need for that and I would suggest uniting these two sections into one, Introduction. Also, be aware that the following chapter is also numbered as section 2 (2. Results and discussion).
  2. This study focuses on Fe1-xAgxCr2S4 (0 < x < 0.2) solid solutions and their magnetic properties. What remains unclear and should be discussed is the effect of the substitution of Fe2+ ions with the Ag+ ions, since they bare different charges. How is charge neutrality achieved, by chromium changing its oxidation state?
  3. On page 2, line 57 the authors state “Previously, our group [24] synthesized samples…”. I would suggest placing the reference at the end of the sentence. The same goes for the  sentence in line 66, “Also, in [25] for Fe1-xAgxCr2S4 (x=0.05-0.15), we measured…”.

Round 2

Reviewer 2 Report

All my questions have been addressed. I think the present manuscript is suitable for publicaton.